# Group-Based Trajectory Modelling of Changes in Mobility over Six Years in Type 2 Diabetes: The Fremantle Diabetes Study Phase II

**DOI:** 10.3390/jcm12134528

**Published:** 2023-07-06

**Authors:** David G. Bruce, Wendy A. Davis, Timothy M. E. Davis

**Affiliations:** Medical School, The University of Western Australia, Fremantle Hospital, Alma Street, Fremantle, WA 6160, Australia; davis.bruce@uwa.edu.au (D.G.B.); wendy.davis@uwa.edu.au (W.A.D.)

**Keywords:** type 2 diabetes, mobility, Timed Up and Go Test, group-based trajectory modelling

## Abstract

To investigate temporal changes in mobility in community-based people with type 2 diabetes, Fremantle Diabetes Study Phase II (FDS2) data were analysed. The baseline assessment included the Timed Up and Go (TUG) test, which was repeated biennially for up to six years. Group-based trajectory modelling (GBTM) identified TUG trajectory groups in participants with ≥2 tests. Independent associates of group membership were assessed using multinomial regression. Of 1551 potential FDS2 participants, 1116 (72.0%; age 64.9 ± 11.0 years, 45.6% female) were included in the modelling. The best-fitting GBTM model identified two groups with linear, minimally changing trajectories (76.2% and 19.4% of participants; baseline TUG times 8 ± 2 and 12 ± 3 s, respectively), and a third (4.5%; baseline TUG 17 ± 5 s) with a TUG that increased over time then fell at Year 6, reflecting participant attrition. Both slower groups were older, more likely to be female, obese, and had greater diabetes-associated complications and comorbidities. Almost one-quarter of the FDS2 cohort had clinically relevant mobility impairment that persisted or worsened over six years, was multifactorial in origin, and was associated with excess late withdrawals and deaths. The TUG may have important clinical utility in assessing mobility and its consequences in adults with type 2 diabetes.

## 1. Introduction

Many adults develop an impaired ability to walk as they age. Such mobility impairments tend to progress over time and eventually become disabling through increased difficulty in performing basic activities of daily living (ADL) [1,2]. They also increase the risk of injurious falls, residential care, and death [3]. A recent meta-analysis found a substantially increased risk of mobility impairment in people with type 2 diabetes [4].

Mobility impairment in type 2 diabetes is usually multifactorial. Documented risk factors include obesity, sedentariness, microvascular complications, macrovascular disease, sarcopenia, lower limb arthritis, and cognitive impairment [5,6,7,8,9]. Several variables are also known to modulate the impact of causal factors and include depression, socioeconomic status, and health literacy [10]. Mobility impairments can develop suddenly or progress gradually [11]. Sudden onset occurs through major illnesses such as stroke or lower limb fracture, while gradual progression may follow a restriction of activity that can be due to many potential causes and which is accelerated by disabling events, including falls and fractures [1,12]. Changes in mobility can be dynamic, and recovery may occur spontaneously or through rehabilitation [13,14]. Attempts at understanding the processes leading to mobility decline are hindered by multiple, potentially interacting, dynamic causative factors.

One potentially fruitful approach is to examine change over time. This could help distinguish groups of individuals at heightened risk of mobility decline who may benefit from preventive intervention. Group-based trajectory modelling (GBTM), employing repeated measures of the variable of interest, provides a relevant empiric methodology that avoids potential pitfalls and limitations from assumed a priori assignment rules [15]. The Timed Up and Go (TUG) test is an objective, validated method for assessing and quantifying mobility performance in healthy and disabled people [16,17] that has been employed in several studies of people with diabetes [18,19,20]. The Fremantle Diabetes Study Phase II (FDS2), a community-based cohort study of people with known diabetes [21], included TUG administered at baseline and biennially for up to six years. The primary aim of the present study was to investigate mobility trajectories in type 2 diabetes using GBTM to empirically define and quantify TUG trajectories in FDS2 participants with type 2 diabetes. A secondary aim was to explore differences in the clinical and demographic characteristics between GBTM-defined trajectory groups.

## 2. Materials and Methods

### 2.1. Study Sample and Approvals

The FDS2 is a prospective, observational study of participants with known diabetes from a post-code-defined urban community of approximately 157,000 people surrounding the port of Fremantle in the state of Western Australia [21]. Those who had participated in the Fremantle Diabetes Study Phase I but were no longer living in the catchment area were also eligible. Sample characteristics, including classification of diabetes type and details of those identified but not recruited, have been described previously [21]. Between 2008 and 2011, 4639 people with diabetes were identified, and 1732 were recruited, including 64 former FDS Phase 1 participants (recruitment 1993-96) who had since moved out of the study area. Of these, 1551 were clinically diagnosed with type 2 diabetes. The South Metropolitan Area Health Service Human Research Ethics Committee approved FDS2 (reference 07/397), and written informed consent was obtained in each case.

### 2.2. Clinical Assessment

All participants underwent a detailed clinical assessment (comprehensive history, physical examination, and special tests to assess cardiovascular risk factors, complications and comorbidities) at entry and then biennially, with multiple attempts to contact participants who did not attend for follow-up in order to minimize attrition [21]. Study questionnaires covered health care utilization; medical conditions; medication use; and socioeconomic, demographic, and lifestyle data. A physical examination was performed by trained Registered Nurses according to a standard protocol. A Body Shape Index (ABSI) was calculated as a more robust index of visceral obesity than Body Mass Index (BMI) [22]. Fasting blood and urine samples were taken for biochemical tests performed using standard automated methods in a single nationally accredited laboratory [23]. Chronic complications were assessed using standard criteria [23,24]. The Charlson Comorbidity Index (CCI) [25] was used to assess comorbidity for the five years before study entry with diabetes and its complications excluded.

The TUG test was performed at each assessment with a possible maximum of four measurements over six years in each participant. The TUG is an integrated test of balance, gait speed, turning ability, and sitting and standing ability that requires individuals to sit, then stand, walk at a comfortable fast and secure pace for 3 m, turn, then return to a sitting position [17]. The test requires a standardized approach, including designated floor space with a mark at 3 m, standard chair height, and a practice session before a single timed assessment with times recorded to the nearest second [17]. The test has no ceiling effect limitation, test results are generally normally distributed, and the TUG is suitable for mobility assessment in healthy as well as impaired individuals [16]. The TUG has high intra- and inter-tester reliability and high construct validity through correlations with gait speed, postural sway, step length, Activities of Daily Living scales, and with identifying people who are at risk of falls. The TUG had moderately high test–retest reliability in one study [16] and high test–retest reliability in a subsequent study [26]. The authors of this latter study stated that a single TUG test is adequate for representing performance, provided the subject understands the different aspects of the test. At baseline and each biennial review, the TUG was performed mid-morning, towards the end of the detailed FDS2 assessment, after fasting blood and urine samples had been collected, self-reported questionnaires had been completed (generally during breakfast), and most physical screening tests had been performed. 

Normal TUG times vary by age, and a recent meta-analysis concluded that individuals with TUG times that exceeded the following age-specific times might warrant interventions for improving balance or mobility: 9.0 s in 60–69-year-olds, 10.2 s in 70–79-year-olds, and 12.7 s in 80–99-year-olds [27]. Alternative TUG thresholds have been recommended, from 12 to 15 s, to define the need for further assessment of fall risk or for rehabilitation purposes [28,29].

### 2.3. Statistical Analysis

The computer packages IBM SPSS Statistics 25 (IBM Corporation, Armonk, NY, USA) and StataSE 15 (StataCorp LP, College Station, TX, USA) were used for statistical analysis. Data from the baseline (Year 0), Year 2, Year 4 and Year 6 assessments were used. Data are presented as proportions, mean ± SD, geometric mean (SD range), or, in the case of variables which did not conform to a normal or log-normal distribution, median and inter-quartile range (IQR). For independent samples, two-sample comparisons were created using Fisher’s exact test for proportions, Student’s *t*-test for normally distributed variables, and Mann–Whitney U-test for non-parametric variables. Comparisons between multiple groups for categorical variables were created using Fisher’s exact or chi-squared tests, comparisons for normally or log-normally distributed continuous variables were created via one-way ANOVA, and comparisons for variables not conforming to normal or log-normal distribution were created via Kruskal–Wallis test. Where the overall trend for these multiple comparisons was statistically significant, post hoc Bonferroni-corrected pairwise comparisons were performed. A two-tailed significance level of *p* < 0.05 was used throughout.

### 2.4. Identification of Trajectory Groups

Group-based trajectory modelling, which employs finite mixture modelling to approximate unknown distributions of trajectories across a study population, was used to identify TUG trajectory groups. Censored normal models were used to estimate trajectories of TUG over six years (four biennial assessments). To assist model selection, the Bayesian Information Criterion (BIC) was utilized to determine the optimum number of groups and their functional form (linear or quadratic) [15]. BIC values balance model fit with model complexity, and the closer the negative BIC value is to zero, the better the fit. Other selection criteria included: (i) adequate numbers of subjects in each group, (ii) distinct trajectories (non-overlapping confidence intervals), (iii) acceptably narrow confidence intervals, (iv) average posterior probabilities of group membership >0.70 for each group, (v) odds of correct classification based on posterior probabilities of group membership >5 for each group, and (vi) close correspondence between each group’s estimated probability and the proportion of participants classified in that group according to the maximum posterior probability assignment rule. 

### 2.5. Characteristics of Trajectory Groups

The bivariable characteristics of the trajectory groups were determined, and multinomial regression was used to identify independent associates of group membership. Clinically relevant and biologically plausible variables were considered for model entry if bivariable *p* < 0.20. Loss to follow-up was quantified overall and by trajectory group, and a logistic regression analysis was undertaken to identify associates of dropout. Since the magnitude of dropout differed by trajectory group, it was adjusted for in the final multinomial models. 

## 3. Results

### 3.1. Participant Sample

Of 1551 FDS2 participants with type 2 diabetes, 1116 (72.0%) had two or more TUG measurements during follow-up. Those with one or no TUG measurement were significantly older (67.7 ± 13.0 versus 64.9 ± 11.0 years, *p* < 0.001), more likely to be female (54.5% versus 45.6%, *p* = 0.002), and had a longer diabetes duration (12.0 [4.0–18.1] versus 8.0 [2.0–15.0] years, *p* < 0.001) compared to those with two or more TUG measurements. 

### 3.2. Trajectory Group Selection and Evaluation

One extreme outlier (TUG time 70 s at Year 6 in a male with incident Parkinson’s disease) was removed because valid GBTM was not possible when he was included. Appendix A shows that the best model for the observed data (i.e., the one with the lowest BIC) was one with three groups, two with linear trajectories, and one with a quadratic trajectory (see Table A1). These have been designated the Fast, Intermediate, and Slow groups based on baseline TUG times and trajectories. Figure 1 illustrates the three trajectories and their narrow associated 95% confidence intervals.

The Fast group was the largest (*n* = 850 (76.2%)) and had a linear trajectory that was generally flat but with a trend to faster performance over time (Fast group: baseline TUG 8 ± 2 s; intercept 8.01 s, linear estimate −0.15 (*p* < 0.001)). The Intermediate group comprised 216 (19.4%) participants with a baseline TUG of 12 ± 3 s and a linear trajectory that was also generally flat but with a gradual slowing over time (intercept 11.52 s, linear estimate +0.13 (*p* = 0.002)). The Slow group comprised 50 participants (4.5%) and had a non-linear trajectory with an initial slow TUG time that increased but then declined (baseline TUG 17 ± 5 s; intercept 16.47 s, linear estimate +3.17 (*p* < 0.001), quadratic estimate −0.47 (*p* < 0.001)). 

Although not all participants had a baseline TUG measure, all those with two or more TUG measures were allocated to one of the three trajectory groups. There was close agreement between observed and predicted TUG values (see Table A2). The average posterior probabilities for the trajectory groups were 0.96 (Fast group), 0.88 (Intermediate group), and 0.97 (Slow group), all greater than the recommended cut-off of 0.7. The odds of correct classification based on the posterior probabilities of group membership were >5 for all groups (Table A3 and Table A4).

### 3.3. Attrition by Trajectory Group

There was a substantial dropout during follow-up that was strongly trajectory-group-dependent (see Table 1). At the Year 4 assessment, 77.9% of the Fast group, 72.2% of the Intermediate group, and 50.0% of the Slow group provided TUG data (*p* < 0.001), and at Year 6, the respective proportions were 69.9%, 50.9%, and 22.0% (*p* < 0.001).

Study attrition can result from several causes, and differential attrition was partially explained by deaths which were also trajectory-group dependent (Fast group 6.9%; Intermediate group 24.5%; Slow group 30.0%; *p* < 0.001). In an exploratory analysis, withdrawal/death was independently associated with older age, higher BMI, higher urinary albumin:creatinine ratio (ACR), estimated glomerular filtration rate (eGFR) <45 mL/min/1.73 m^2^, and the presence of peripheral arterial disease (*p* < 0.05).

### 3.4. Baseline Characteristics of Trajectory Groups

There were significant differences in baseline characteristics between the TUG trajectory groups (see Table 2).

Compared to the Fast group (mean age 62.6 ± 10.6 years, 57.3% male), those in the Intermediate and Slow groups were significantly older (71.6 ± 8.9 and 74.9 ± 8.2 years, respectively) and more likely to be female (52.3% and 66.0%, respectively). There were differences in education, English fluency, and time watching television. Self-reported problems with both mobility and ADL were uncommon in the Fast group (reported by 6.6% and 3.4%, respectively), more common in the Intermediate group (reported by 27.5% and 9.1%, respectively), and most frequent in the Slow group (reported by 71.1% and 20.0%, respectively). Variables associated with Intermediate and Slow group membership included obesity, diastolic blood pressure, diabetes duration, insulin therapy, albuminuria, eGFR, peripheral neuropathy, coronary heart disease, cerebrovascular disease, and other non-diabetes-associated comorbidity. 

In multinomial regression, older age, female sex, higher BMI, and a history of cerebrovascular disease were significantly associated with membership in both the Intermediate and Slow groups (Table 3). Additional associates were urinary albumin:creatinine ratio and the CCI with membership of the Intermediate Group and lower diastolic blood pressure and coronary heart disease with membership of the Slow group. The models were adjusted for dropouts given the independent association with group membership.

## 4. Discussion

In the present study, three distinct TUG trajectory groups of community-based people with type 2 diabetes were defined using GBTM, one with normal TUG times and two with TUG times, indicating different degrees of mobility impairment [27]. The group with the slowest TUG time at baseline had a complex trajectory with worsening TUG times over the next four years, followed by an apparent improvement that was probably explained by substantial attrition from the group by the end of the follow-up. The other trajectory groups exhibited only minor changes during follow-up, indicating little or no clinically relevant change in mobility over six years. Overall, almost 25% of the study participants were members of the two groups with TUG times indicating some degree of mobility impairment. These data indicate that clinically relevant mobility impairments are common in type 2 diabetes and that the TUG test is likely to be useful in differentiating individuals likely to have progressive problems from those likely to have stable mobility for up to six years.

The Slow group had a mean baseline TUG time of 17 s and an initially increasing trajectory, indicating progressively declining mobility. A TUG time of 17 s is well above age-matched norms [27], is consistent with the presence of physical frailty [30], and increases the risk of falls and disability [25,26]. This group was the smallest and had the highest proportional withdrawal and death rates. Consequently, the apparent improvement in mobility seen at six years is likely explained by attrition bias due to the differential loss of those with the worst mobility. The Intermediate group had an average baseline TUG time of 12 s, which is close to the threshold for clinically relevant impairment [28,29]. This indicates that a proportion of the group may also have experienced mobility-related adverse consequences [3], as evidenced by the increased rate of withdrawals and deaths compared with the Fast group.

There were demographic and clinical differences between the three GBTM-defined groups. The group with normal mobility was the youngest, with an average age of 63 years, whereas both mobility-impaired groups were in their early to mid-70s. These latter groups also had a longer duration of diabetes and more complications, comorbidities, and self-reported mobility limitations and disabilities. Membership of both slow TUG trajectory groups was multifactorial, and independent associates included age, female sex, and greater obesity, variables previously associated with mobility impairments in older adults [6,31,32]. Albuminuria was the only microvascular association, possibly explained by the known association with sarcopenia [9], a variable that was not measured in this study. Macrovascular disease, especially cerebrovascular disease, and a higher burden of non-diabetes comorbidities were also important associated variables. 

The results of the present study are consistent with previous studies demonstrating high rates of mobility limitation in people with type 2 diabetes that progress over time [33,34,35]. To our knowledge, no previous studies have explored mobility trajectories in type 2 diabetes. A study of disability trajectories, comparable given the close association between mobility and disability [36], had broadly similar findings to the present study in older patients with self-reported diabetes [37]. Three disability trajectory groups were defined using latent class analysis; one group remained non-disabled, and two groups had a mild and more severe disability at baseline that steadily progressed over eight years [37]. 

Our study has clinical implications. Exercise-based interventions have been shown to be effective in diabetes and can lead to improvements in mobility [38,39,40] and reduce the risk of falls [40]. Our data suggest that a mobility assessment can be useful in risk prediction in older adults with type 2 diabetes. The age differences between the trajectory groups suggest that mobility impairments first started at an earlier age in both groups with mobility impairments. A screening assessment at or around age 65 years appears justified based on our data. While several validated mobility tests are available for screening purposes [41], the TUG test is easy to administer, analyse, and interpret; requires little equipment; and can be conducted in most clinical settings without training in less than 5 min [17,41]. A slow TUG time would indicate the need for a more comprehensive assessment generally conducted by a multidisciplinary team [29]. A normal time would reassure the clinician and provide an assessment likely to be valid for several years in the absence of any intercurrent disabling events. The TUG times seen in the intermediate group may also be clinically relevant as they may indicate a milder degree of impairment that could act as a barrier to healthy activity levels as an integral part of diabetes self-management [42]. 

The present study has several strengths, including the size and representative nature of the FDS2 cohort, the comprehensive nature of the clinical assessment, and the use of group-based trajectory analysis to handle longitudinal assessments of mobility. Such an analysis helps reduce random variations related to single measurements and provides higher accuracy of derived groupings. The limitations of this study included recruitment and retainment biases inherent in all cohort studies, where less-impaired and healthier individuals may be more likely to participate and return for follow-up assessments. The assessment of potential associates, which was reasonably comprehensive, did not include a detailed assessment of neuropathy severity, which might explain the lack of association with mobility or sarcopenia, which is an important determinant of mobility impairment [9]. The derived models included the group with a small sample size, which limits the power of the exploratory analysis, especially given the rate of dropouts from the study and this group in particular. 

## 5. Conclusions

Group-based trajectory modelling revealed that 24% of adults with type 2 diabetes in our cohort had impairments in mobility that were persistent or worsened over six years of follow-up. The cause of mobility impairment was multifactorial, but older people, particularly females and those with greater obesity, were most at risk. Assessing mobility with the TUG may be a useful clinical adjunct in the assessment of adults with type 2 diabetes.

## Figures and Tables

**Figure 1 jcm-12-04528-f001:**
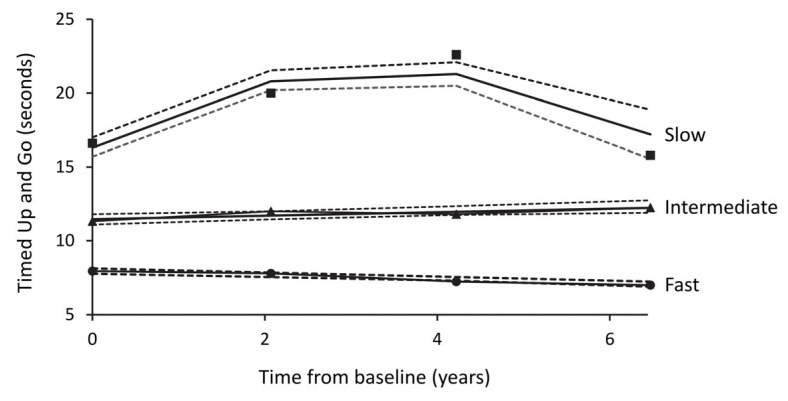
Three Timed Up and Go trajectory groups that best fit the observed data derived from group-based trajectory modelling.

**Table 1 jcm-12-04528-t001:** Numbers of participants and cumulative deaths at each assessment in the three Timed Up and Go trajectory groups.

	Fast	Intermediate	Slow
Total number per group	850	216	50
Number (%) assessed at baseline	836 (98.4)	211 (97.7)	47 (94.0)
Number (%) assessed at Year 2	820 (96.5)	202 (93.5)	44 (88.0)
Number (%) assessed at Year 4	662 (77.9)	156 (72.2)	25 (50.0)
Number (%) assessed at Year 6	594 (69.9)	110 (50.9)	11 (22.0)
Number (%) deaths by end Year 2 follow-up *	23 (2.7)	14 (6.5)	4 (8.0)
Number (%) deaths by end Year 4 follow-up ^†^	39 (4.6)	32 (14.8)	9 (18.0)
Number (%) deaths by end Year 6 follow-up ^‡^	59 (6.9)	53 (24.5)	15 (30.0)

* All had a Year 2 TUG; ^†^ Year 4 TUG assessments were conducted in 12, 14, and 1 participants in the Fast, Intermediate and Slow groups, respectively, before death prior to Year 4; ^‡^ Year 6 TUG assessments were conducted in 6, 4, and 2 participants in the Fast, Intermediate, and Slow groups, respectively, before death occurred prior to Year 6 closeout.

**Table 2 jcm-12-04528-t002:** Baseline characteristics by derived Timed Up and Go trajectory groups.

	Fast	Intermediate	Slow	*p*-Value
Number (%)	850 (76.2)	216 (19.4)	50 (4.5)	
Age (years)	62.6 ± 10.6	71.6 ± 8.9 ***	74.9 ± 8.2 ***	<0.001
Sex (% male)	57.3	47.7 *	34.0 **	0.001
Ethnic background (%):				0.286
Anglo-Celt	55.3	61.6	58.0	
Southern European	10.9	13.0	18.0	
Other European	7.9	5.6	8.0	
Asian	5.1	2.8	0	
Aboriginal	4.5	2.3	4.0	
Other	16.4	14.8	12.0	
Not fluent in English (%)	6.7	12.0 *	24.0 ***	<0.001
Secondary education (%)	93.8	82.1 ***	63.3 ***^,†^	<0.001
Smoking status (%):				0.55
Never	40.9	45.6	48.0	
Ex-smoker	50.4	47.4	42.0	
Current smoker	8.7	7.0	10.0	
Alcohol (drinks/day)	0.3 [0–1.5]	0.1 [0–0.8] ***	0 [0–0.3] **	<0.001
TV viewing ≥21 h/week (%)	32.7	43.1 *	43.5	0.010
Any mobility limitation (%)	6.6	27.5 ***	71.1 ***^,†††^	<0.001
Any problem with ADLs (%)	3.4	9.1 **	20.0 ***	<0.001
BMI (kg/m^2^)	30.9 ± 5.7	32.6 ± 6.0 ***	33.3 ± 6.8 *	<0.001
Obesity by waist circumference ^a^ (%)	68.2	76.9 *	94.0 ***^,†^	<0.001
ABSI (m^11/6^ kg^−2/3^)	0.081 ± 0.005	0.082 ± 0.005 **	0.084 ± 0.005 ***	<0.001
Heart rate (beats/min)	68 ± 11	70 ± 14	72 ± 14	0.014
Systolic blood pressure (mm Hg)	145 ± 21	148 ± 22	145 ± 22	0.082
Diastolic blood pressure (mm Hg)	82 ± 12	79 ± 13 **	74 ± 10 ***^,†^	<0.001
Taking antihypertensives (%)	71.2	85.6 ***	76.0	<0.001
Age, diabetes diagnosis (yrs)	54.2 ± 11.2	59.6 ± 12.0 ***	61.1 ± 12.0 ***	<0.001
Diabetes duration (yrs)	6.0 [2.0–14.0]	12.0 [5.0–16.6] ***	14.7 [4.8–20.8] ***	<0.001
Diabetes treatment (%):			**	0.001
Diet	27.3	20.4	16.0	
Non-insulin medications	53.6	54.6	42.0	
Insulin alone	4.0	4.2	16.0	
Insulin ± other agents	15.1	20.8	26.0	
Fasting serum glucose (mmol/L)	7.2 [6.2–8.7]	7.1 [6.1–8.7]	7.5 [6.4–8.9]	0.535
HbA_1c_ (%)	6.8 [6.1–7.7]	6.9 [6.3–7.6]	7.1 [6.6–8.1]	0.114
HbA_1c_ (mmol/mol)	51 [43–61]	52 [45–60]	54 [49–65]	0.114
Serum total cholesterol (mmol/L)	4.4 ± 1.1	4.3 ± 1.1	4.4 ± 2.1	0.800
Serum HDL cholesterol (mmol/L)	1.22 ± 0.32	1.29 ± 0.32 *	1.21 ± 0.30	0.027
Serum triglycerides (mmol/L)	1.5 (0.9–2.5)	1.5 (0.9–2.5)	1.6 (1.0–2.8)	0.501
Lipid-lowering medication (%)	68.2	73.6	76.0	0.188
Aspirin use (%)	35.9	43.3	44.0	0.086
Urinary ACR (mg/mmol)	2.7 (0.8–9.3)	4.6 (1.3–16.6) ***	4.6 (1.2–17.2) **	<0.001
eGFR categories ^b^ (%):		***	***	<0.001
≥90 mL/min/1.73 m^2^	44.9	24.8	12.0	
60–89 mL/min/1.73 m^2^	45.8	50.9	52.0	
45–59 mL/min/1.73 m^2^	5.7	12.6	18.0	
30–44 mL/min/1.73 m^2^	2.8	8.4	16.0	
<30 mL/min/1.73 m^2^	0.8	3.3	2.0	
Any retinopathy (%)	34.5	39.9	36.7	0.320
Peripheral neuropathy (%)	52.4	70.8 ***	82.0 ***	<0.001
Peripheral artery disease (%)	16.4	25.9 **	40.0 ***	<0.001
Coronary heart disease (%)	21.9	37.5 ***	56.0 ***	<0.001
Cerebrovascular disease (%)	4.2	14.4 ***	18.0 **	<0.001
CCI ^c^ (%):		***	*	<0.001
0	83.2	65.7	68.0	
1 or 2	13.5	21.3	20.0	
≥3	3.3	13.0	12.0	

* *p* < 0.05 versus Fast group; ** *p* < 0.01 versus Fast group; *** *p* < 0.001 versus Fast group; ^†^
*p* < 0.05 versus Intermediate group; ^†††^
*p* < 0.001 versus Intermediate group; ^a^ waist circumference ≥102 cm in men and ≥88 cm in women; ^b^ estimated glomerular filtration rate from the Chronic Kidney Disease Epidemiology Collaboration equation; ^c^ CCI derived for the 5 years before study entry.

**Table 3 jcm-12-04528-t003:** Multinomial logistic regression models of influential factors on Timed Up and Go trajectory group membership relative to the Fast group in 1116 participants with type 2 diabetes. RRR = relative risk ratio. All variables in Table 2 were considered for entry. Models were adjusted for dropout probabilities and all factors in the table below.

	Intermediate Group		Slow Group	
Factors	RRR	95% CI	RRR	95% CI
Age (increase of 1 year)	1.12	1.09, 1.15	1.19	1.13, 1.25
Male sex	0.61	0.42, 0.88	0.31	0.15, 0.63
BMI (increase of 1 kg/m^2^)	1.13	1.09, 1.16	1.18	1.11, 1.25
Diastolic blood pressure (increase of 1 mm Hg)	0.99	0.97, 1.01	0.96	0.93, 0.98
Ln(urinary ACR (mg/mmol))	1.20	1.05, 1.37	1.06	0.80, 1.42
Coronary heart disease	1.14	0.78, 1.67	2.65	1.36, 5.17
Cerebrovascular disease	2.43	1.33, 4.46	3.79	1.40, 10.2
CCI ≥ 3	3.08	1.52, 6.20	2.57	0.79, 8.30

## Data Availability

Some outcome data supporting the findings of this study are available from the Western Australian Department of Health, but restrictions apply to the availability of these data, which were used under strict conditions of confidentiality for the current study, and so are not publicly available. Data are, however, available from the authors upon reasonable request and with permission of the Western Australian Department of Health.

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
