# Peer review of "Group-Based Trajectory Modelling of Changes in Mobility over Six Years in Type 2 Diabetes: The Fremantle Diabetes Study Phase II"

_jcm, 2023, doi:10.3390/jcm12134528_

Round 1
Reviewer 1 Report
Simplistic study with limited measures presented. The strength is the longitudinal design, the statistical analysis, and the manuscript is well written. However, the paper would benefit from including an analysis of the additional physical assessments that were conducted during the larger study.
Information on the test-retest reliability of the TUG and on the time of assessments are missing.
The study would have benefited from more sensitive and in-depth gait and balance measures (e.g. gait analysis systems and force plates).
As presented, the manuscript lacks innovation and does not advance prior knowledge…
Author Response
Simplistic study with limited measures presented. The strength is the longitudinal design, the statistical analysis, and the manuscript is well written. However, the paper would benefit from including an analysis of the additional physical assessments that were conducted during the larger study.
Response: Regarding the use of the terms ‘simplistic’ and ‘limited’, the FDS2 covered all aspects of diabetes of which mobility was one. We restricted our assessments to usual care measures, in part because of the time constraints and burden on participants but also because our findings could be interpreted in the light of the provision of routine management. The TUG test is a highly recommended, rapid screening test for people with mobility limitations that can be used in clinical areas including primary care and diabetes clinics. It is important to understand how this test performs in the long-term, especially in people in mid-life approaching older age and at risk for clinically important mobility problems.
Table 2 presents relevant physical measurements and questionnaire responses undertaken at the baseline assessment. Many variables are colinear and so we undertook multivariable analysis to identify the key drivers of TUG Trajectory membership (Table 3). In this way, a full range of relevant candidate variables were included in the analysis. We have further clarified this in the caption to Table 3 which now includes the statement: “All variables in Table 2 were considered for entry. Models were adjusted for drop-out probabilities and all factors in the table below.”
Information on the test-retest reliability of the TUG …
Response: The TUG has been reported to have high intra- and inter-tester reliability, high construct validity through correlations with gait speed, postural sway, step length, ADL scales, and with identifying people who fall. The test had moderately high test-retest reliability in one study (reference #16 in the paper) and high test-retest reliability in a subsequent study (Steffen TM et al Age- and gender-related test performance in community-dwelling elderly people: Six-minute walk test, Berg balance scale, time up & go test, and gait speed. Physical Therapy 82: 128-137, 2002). The authors stated that a single trial of the TUG test is adequate for representing performance provided the subject understands the different aspects of the test. The Methods section has been amended to include these details and the additional reference.
….and on the time of assessments are missing.
Response: At baseline and each biennial assessment, the TUG was performed mid-morning, towards the end of the physical assessment which occurred after fasting bloods and urine were provided and breakfast consumed. Self-reported questionnaires were generally completed during the breakfast break. This is now specified in the Methods section: “At baseline and each biennial review, the TUG was performed mid-morning, towards the end of the detailed FDS2 assessment, after fasting blood and urine samples had been collected, self-reported questionnaires had been completed (generally during breakfast), and most physical screening tests had been performed.“
The study would have benefited from more sensitive and in-depth gait and balance measures (e.g. gait analysis systems and force plates).
Response: Please see Response above for the reasons why the other more specialised physical measures were not included. Force plate and gait analysis systems remain largely confined to human movement research and are rarely utilised in usual clinical practice. Clinically the challenge is to identify people in the early stages of mobility decline or at risk of physical disability or falls who can benefit from physiotherapy or similar rehabilitation efforts. The TUG test is a useful screening test to identify potential candidates. The TUG test can thus obviate the need for more sophisticated, relatively expensive and time-consuming tests.
As presented, the manuscript lacks innovation and does not advance prior knowledge…
Response: To the best of our knowledge, this is the first study to measure TUG performance longitudinally in type 2 diabetes. It is also the first to use group based trajectory modelling in any mobility measure over a prolonged time period. We believe that this study has important clinical implications and supports the wider use of the TUG test (or of other similar simple-to-perform mobility screening tests) in routine clinical practice. The trajectory analysis implies that slow TUG times indicate the need for further assessment or surveillance of mobility ability in some people. In contrast, those with fast TUG times can be expected to remain normally mobile provided they do not experience a clinically relevant disorder or event.
Reviewer 2 Report
In this study the authors investigate mobility trajectories in FDS participants with Type 2 diabetes. The study is valid and well conducted. Given the extensive différences across TUG categories in clinical characteristics, as shown in table 2, could the authors expand or comment on other comorbidities that could be driving the observed differences?
Author Response
In this study the authors investigate mobility trajectories in FDS participants with Type 2 diabetes. The study is valid and well conducted. Given the extensive différences across TUG categories in clinical characteristics, as shown in table 2, could the authors expand or comment on other comorbidities that could be driving the observed differences?
Response: Table 2 presents the bivariable associations of Trajectory Group membership, whereas Table 3 shows the independent associations derived from multivariable analysis. We describe in the Results that the presence of cerebrovascular disease is associated with membership of both the intermediate and slow groups, coronary heart disease with membership of the slow group, and a Charlson Comorbidity Index (CCI) ≥3 with the intermediate group. We have described the CCI in the Methods section, but it includes a broad spectrum of non-diabetes-specific comorbidities (cancer, heart failure, chronic pulmonary disease, dementia, HIV/AIDS, paralysis, peptic ulcer disease, rheumatic disease, liver disease). We have not added these conditions individually to the analysis since we were focussed on diabetes-related reasons (including cerebrovascular disease, coronary heart disease, peripheral arterial disease, and renal disease). The higher the CCI, the greater the burden of non-diabetes-specific comorbidities.